# A Quick Guide to CAF Subtypes in Pancreatic Cancer

**DOI:** 10.3390/cancers15092614

**Published:** 2023-05-04

**Authors:** Anna Brichkina, Pierfrancesco Polo, Shrey Dharamvir Sharma, Nico Visestamkul, Matthias Lauth

**Affiliations:** Center for Tumor and Immune Biology, Clinics for Gastroenterology, Endocrinology and Metabolism, Philipps University Marburg, Hans-Meerwein-Str. 3, 35043 Marburg, Germany; anna.brichkina@staff.uni-marburg.de (A.B.); polo@staff.uni-marburg.de (P.P.); sharmash@staff.uni-marburg.de (S.D.S.); visestan@staff.uni-marburg.de (N.V.)

**Keywords:** tumor stroma, pancreatic cancer, cancer-associated fibroblast, CAF, stellate cell, ECM, TME

## Abstract

**Simple Summary:**

Solid cancers are composed of a mixture of various cell types. Tumor cells are among this agglomeration and interact with other components through cell-cell contacts and released factors. Pancreatic cancer is particularly rich in additional cell types which are instructed by neighboring cancer cells to behave in a cancer-promoting manner. One of the abundant cell types in pancreatic cancer are the so called cancer-associated fibroblasts (CAFs). CAFs exist as different subsets with distinct properties which can be tumor growth stimulating or even be tumor restraining. This review describes the various CAF subpopulations, their molecular discrimination, and the resulting functional impact of these cells on pancreatic cancer development and therapy approaches.

**Abstract:**

Pancreatic cancer represents one of the most desmoplastic malignancies and is characterized by an extensive deposition of extracellular matrix. The latter is provided by activated cancer-associated fibroblasts (CAFs), which are abundant cells in the pancreatic tumor microenvironment. Many recent studies have made it clear that CAFs are not a singular cellular entity but represent a multitude of potentially dynamic subgroups that affect tumor biology at several levels. As mentioned before, CAFs significantly contribute to the fibrotic reaction and the biomechanical properties of the tumor, but they can also modulate the local immune environment and the response to targeted, chemo or radiotherapy. As the number of known and emerging CAF subgroups is steadily increasing, it is becoming increasingly difficult to keep up with these developments and to clearly discriminate the cellular subsets identified so far. This review aims to provide a helpful overview that enables readers to quickly familiarize themselves with field of CAF heterogeneity and to grasp the phenotypic, functional and therapeutic distinctions of the various stromal subpopulations.

## 1. Introduction

Humans are multicellular organisms, and each cell is allocated a specific task to provide selective functionality to the entire body. In order to ensure proper interplay, several cell-intrinsic and cell-extrinsic mechanisms have evolved to remove or control cells which have gained non-physiological properties, such as malignancy. Solid tumors are embedded in the tissue from which they originate, and normal microenvironmental structures often possess a surprising capacity to restrain tumor growth [1]. However, over time, cancer cells acquire control over their surrounding neighbors, essentially reprogramming them into cancer-associated counterparts that foster tumor growth [2,3]. These processes, which mimic tissue repair and wound healing, almost inevitably involve mesenchymal cells in the form of fibroblasts [4]. This review aims to outline the current knowledge on CAF heterogeneity encountered in solid cancer (with pancreatic cancer being in the focus) and the functional consequences that these individual subpopulations exert on tumor development, maintenance and therapy outcome.

## 2. A Large Signaling Network Activates Fibroblasts

Fibroblasts are ubiquitously encountered constituents of the tumor microenvironment in solid cancers. Since their normal role in physiology is repair and homeostasis, they can be found in virtually all tissues, where they often stay quiescent and do not stand in the functional foreground [5]. This holds true for other independent lines of resident mesenchymal cells, such as the stellate cells of the liver and pancreas. This quiescent scenario changes abruptly when signals associated with tissue damage and repair are induced. Many such signals have been delineated and include transforming growth factor β (TGF-β), Hedgehog (Hh), Wnt, platelet-derived growth factor (PDGF), interleukins or tumor necrosis factor (TNF) [5]. Additionally, mechanical cues can instruct quiescent fibroblasts to become activated through the Hippo-YAP/TAZ and MRTF/SRF pathways [6,7].

Of these, the prototypic TGF-β ligands are the best understood members of extracellular ligands, which promote the transition from resident fibroblasts to wound-healing myofibroblastic CAFs by activating SMAD-dependent transcription (myCAFs, discussed later) [8]. Much of the older literature focuses on TGF-β and myCAFs in wound healing, fibrosis and cancer, as additional CAF subtypes were unknown until recently. Moreover, several general mediators of CAF activation have been identified, which function as nodes in an intricate signaling network, often in a self-amplifying manner. For instance, heat-shock factor 1 (HSF1) has been delineated as important promoter of overall CAF activation in several cancer entities, including PDAC [9,10,11]. HSF1 performs a transcriptional program distinct from classical heat stress and induces the expression of TGF-β ligands and CXCL12, supporting an autocrine signaling loop of CAF activation [9]. Another HSF1 target gene is *DKK3*, leading to the promotion of Wnt signaling and the subsequent activation of the Hippo pathway members YAP/TAZ [6,12]. The transcriptional co-activators YAP/TAZ are critical mediators of the CAF state in cancer, including PDAC [6,13]. Similar processes apply to disease-related fibroblasts, where the key fibrotic regulator PU.1 drives the production of the YAP/TAZ interactor TEAD1 and the expression of smooth muscle actin and collagen: two cardinal markers of myofibroblasts [14].

As producers of collagenous ECM, CAFs are embedded in a stiff environment. Tissue stiffness further stimulates YAP/TAZ activity through integrin-induced remodeling of the actin cytoskeleton [6,15]. Other modulators of the actin network, such as the actin-severing and capping protein gelsolin, can also impact the process of fibroblast activation [16]. Along these lines, mechano-sensitive focal adhesion kinase (FAK), which lies downstream of integrin receptors, also contributes to fibroblast activation, ECM deposition, fibrosis and immune suppression [17]. However, the precise role of FAK in pancreatic CAFs might require additional investigations, as another study ascribed a tumor-suppressive function to stromal FAK [18].

As stromal mesenchymal cells transition from a quiescent to an activated state with strongly distinct characteristics, it is not surprising that epigenetic changes play important roles in this transdifferentiation. The processes involved are not fully elucidated, but some studies shed light on the underlying molecular steps. For instance, DNA methylation, as regulated through DNMT3B and DNMT1, controls CAF features in human lung and head–neck carcinomas [19]. Along these lines of research, the cellular availability of methyl groups to epigenetically modify DNA or histones is subject to negative regulation by nicotinamide N-methyl transferase (NNMT). High stromal NNMT levels have been found in breast, colon and ovarian cancer, and its activity is critically required for CAF function. NNMT reduces the overall available methyl group pool, thereby favoring the hypomethylation of repressive chromatin/DNA marks and the induction of CAF-related gene transcription [20].

Although stromal fibroblasts are considered not to harbor major driver gene mutations, they may acquire changes in basal gene expression that predispose them to activation by environmentally provided signaling cues. For example, stromal downregulation of the p62/SQSTM1 protein results in the activation of fibroblasts and hepatic stellate cells (HSCs), the latter being functionally similar to pancreatic stellate cells (PSCs) [21,22]. The p62 adapter typically binds to the Vitamin D receptor (VDR) and promotes the heterodimerization of VDR and the Retinoic X receptor (RXR). As VDR-induced target genes drive stellate cells into quiescence [23], loss of p62 generally maintains an activated state. Furthermore, P62 plays a role in autophagy, which has been deemed critical for CAF activation [24,25]. Interestingly, p62 can be transcriptionally suppressed by lactate, an abundant metabolite in the TME of several cancers [26], or by hypoxia [25], providing mechanistic explanations of how the local microenvironment can generally modulate and stimulate overall CAF activity.

## 3. The Origin of CAFs

Several cell populations have been suggested to give rise to pancreatic CAFs, such as resident fibroblasts or stellate cells (PSCs) [27,28], bone marrow- or adipose tissue-derived mesenchymal stem cells (MSCs) or monocytes [29,30], pericytes that have undergone an endothelial-to-mesenchymal transition (EndMT) [31], or cancer cells that have completed an epithelial-to-mesenchymal transition (EMT) [32]. However, most studies favor the view that the majority of CAFs derive from local sources within the pancreas. As a result of the highly complex interplay between different cell types in the tumor microenvironment (TME), CAF heterogeneity is created. Nevertheless, novel studies applying lineage tracing approaches indeed suggest a pre-existing fibroblast heterogeneity that would ultimately lead to distinct CAF subpopulations in the context of disease (see section on HOXB6^+^ CAFs below). This pre-existing heterogeneity might already be established during embryogenesis [33]. Thus, it is currently debated whether CAF heterogeneity is primarily formed through embryonic pools with defined cellular trajectories or whether particular cell states derive from locally present signaling molecules such as TGF-β or IL-1α, or whether both scenarios are true.

## 4. Heterogeneity of Cancer-Associated Fibroblasts in PDAC

Pancreatic ductal adenocarcinoma (PDAC) displays a highly abundant fibrous inflammatory process called desmoplasia, which leads to poor responses of PDAC to conventional chemotherapy and immunotherapy. The cellular components of the desmoplastic stroma in PDAC are composed primarily of pancreatic stellate cells (PSC) and cancer-associated fibroblasts (CAFs) [34,35]. In the final tumor, these two cell populations are difficult to distinguish as PSCs also differentiate into CAFs during cancer development (see section on “origin of CAFs”). Hence, the term *CAFs* is commonly used in the literature and in this review to describe all activated fibroblastic cells in the TME, regardless of their origin.

CAFs have been shown to secrete growth factors, inflammatory ligands and extracellular matrix proteins (ECM) involved in cancer cell proliferation, therapy resistance and immune exclusion. Early in vivo and in vitro studies postulated that the predominant role of CAFs is to enhance cancer cell proliferation and invasion in PDAC and other cancers [36,37,38,39]. Several studies have implicated the fibroblastic stroma and ECM as a physical barrier to the delivery of cytotoxic chemotherapies to the peritumoral milieu [34,40,41,42]. These observations have led to the paradigm that tumor stroma supports and promotes the growth of cancer. Fibroblasts are therefore a key determinant in the malignant progression of cancer and represent an important target for cancer therapies [3]. Nonetheless, although preclinical models have demonstrated the benefit of stromal depletion, for example through blockade of paracrine Hedgehog signaling in accentuating drug delivery [40], subsequent clinical trials targeting stromal fibroblasts in human PDAC resulted in a paradoxically accelerated disease progression [43,44]. Advances in genetically engineered mouse models of PDAC, single-cell RNA sequencing, and complex co-culture experiments have highlighted profound heterogeneity of cancer-associated fibroblasts, suggesting that stromal fibroblasts in PDAC might exhibit context-dependent functions, imparting tumor-promoting and -restraining influences [45,46,47,48]. The group of David Tuveson made a groundbreaking discovery in 2017, asserting that CAFs in PDAC exhibit either a myofibroblastic (myCAF) or an inflammatory (iCAF) phenotype [49]. This broad classification of myCAFs and iCAFs was later confirmed in human and mouse tissue by means of follow-up single cell RNA sequencing approaches [47,48,50]. It is now established that these two subtypes represent the two largest CAF fractions in PDAC. Across various cancer types, myofibroblastic CAFs are associated with an ECM signature and contain at least some tumor-restricting function, whereas non-myofibroblastic CAFs are generally characterized by a secretory, inflammatory phenotype (iCAFs) involved in immune suppression, further supporting tumor survival and growth. Overall, the existence of both myofibroblastic and inflammatory populations of CAFs seems to be consistent among different cancer types [47,51,52,53,54]. Appendix A summarizes the currently identified CAF subtypes and classifies them into pro- and anti-tumorigenic groups (and in CAFs of unknown function) in order to allow for a quick overview. Figure 1, Figure 2 and Figure 3 represent this classification graphically.

## 5. Myofibroblastic CAFs (myCAFs)

Although CAFs have traditionally been considered stromal cells with pro-tumorigenic functions, several clinical and preclinical efforts to target or deplete CAFs in PDAC have revealed facilitated cancer progression, highlighting the heterogeneity of CAFs and the existence of certain fibroblasts endowed with anti-tumor properties [49,55,56,57]. Currently, myofibroblasts are typically associated with tumor-restraining properties [49]. Nevertheless, it should be mentioned that there is also considerable evidence for myCAFs possessing tumor-promoting capabilities, for instance through ECM deposition and the subsequent increase in tissue stiffness, through paracrine metabolic support and by impacting immune cell motility [58,59,60,61,62]. Thus, the interplay within pancreatic tumors is very complex, and its outcome might be context-dependent and influenced by the experimental framework and/or read-outs applied in some cases.

Myofibroblastic CAFs are located closely to tumor cells, express a high level of alpha smooth muscle Actin (α-SMA) and are stimulated by TGF-β. The high expression level of α-SMA is a well-established marker of myofibroblasts used to define myCAFs on histological sections, in in vitro assays or transcriptomic analyses. Disruption of proper differentiation of pancreatic stellate cells (PSC) into α-SMA^+^ myofibroblasts, in part regulated by Rho effector protein kinase N2 (PKN2), could shift the activation towards inflammatory CAFs and promote pancreatic tumor invasion [63]. The distinct feature of myCAFs is the production of a large amount of extracellular matrix in the tumor stroma, with the most abundant components of ECM produced by CAFs being collagens and hyaluronan. Advances in genetic manipulations in mice by tracking and ablating specific CAF populations has defined their key functions and their role in PDAC development. Özdemir et al. used a genetic approach to deplete α-SMA^+^ myofibroblasts in mice selectively in the early and late stages of PDAC [55]. MyCAF depletion reduced collagen I content and altered ECM organization, with a significantly decreased tumor tissue stiffness leading to aggressive tumors with diminished animal survival. Although myofibroblast-depleted tumors did not respond to gemcitabine, they were sensitized to anti-CTLA4 immunotherapy, opening potential therapeutic avenues. Similarly, genetic ablation or pharmacological inhibition of Sonic Hedgehog (Shh) signaling in mouse PDAC dramatically reduced the α-SMA^+^ stromal content and increased vascularity. This resulted in more aggressive tumors exhibiting an undifferentiated histology, heightened proliferation and increased metastatic potential [56,64]. These results are consistent with studies showing that lower stromal density correlates with a worse prognosis of human PDAC, often due to a higher incidence of metastases [55,65,66,67,68,69]. All these data demonstrate an important protective role of myofibroblasts and myCAF-secreted collagens and ECM in PDAC and urge caution in clinically deploying non-selective stromal depletion strategies in PDAC. It appears that the desmoplastic reaction represents a host defense mechanism, similar to wound healing and tissue regeneration, to repair or hopefully impede the conversion of a neoplastic lesion into invasive carcinoma.

## 6. myCAFs: Complexities around Collagen

The PDAC ECM contains combinations of type I, III, IV, V and XV collagens; collagen type I and collagen type III collectively account for approximately 90% of the total collagen mass in PDAC [70]. Collagens are large macromolecules that upon deposition into the ECM can form fibrillar structures which provide a mechanical framework to the TME. Collagens are involved in cell–stromal interactions through different receptor families such as integrins, discoidin domain receptors (DDR), glycoprotein VI and leukocyte-associated immunoglobulin-like receptor-1 (LAIR-1) [71]. Activated myCAFs are major contributors of collagen I to the PDAC stroma, but tumor cells are able to produce ECM as well [70]. Recently, it was demonstrated that the specific deletion of collagen I in α-SMA^+^ cells accelerated pancreatic tumor progression with diminished overall survival [65]. Deletion of collagen I in hepatic stellate cells also facilitated metastatic spread and homing of pancreatic cancer cells into the liver [72]. Interestingly, specific deletion of collagen I in Fsp1^+^ stromal cells did not impact PDAC progression, again underscoring the importance of CAF heterogeneity and suggesting that the collagen contribution of Fsp^+^ CAFs is pathophysiologically irrelevant [65]. Mechanistically, the α-SMA^+^-selective depletion of collagen I was associated with the recruitment and polarization of CD206^+^F4/80^+^Arg1^+^ MDSCs and impaired recruitment and activation of T and B lymphocytes, unravelling a fundamental role of myofibroblast-derived collagen I in the regulation of immune responses restraining PDAC progression.

Ninety percent of stromal collagen I is produced by CAFs; however, the remaining 10% are deposited by other cells in the TME, including pancreatic cancer cells [70]. Importantly, collagen I produced by CAFs is built up as heterotrimers (encoded by *COL1A1* and *COL1A2*), whereas pancreatic cancer cells produce COL1A1 homotrimers due to promoter hypermethylation of the *COL1A2* gene [73]. Specific ablation of *Col1a1* expression in epithelial cancer cells significantly delayed PDAC development, resulting in an increase in overall survival associated with enhanced T cell infiltration and higher sensitivity to anti-PD-1 immunotherapy. Surprisingly, loss of collagen I homotrimers correlated with a shift from an anaerobic to a microaerophilic tumor microbiome and a normalization of the gut microbiome in PDAC mice [73]. Additionally, the C-terminal pro-domains of fibrillar collagens are partially uncleaved in PDAC ECM, suggesting reduced procollagen C-proteinase activity. BMP1 (bone morphogenetic protein 1), the proteinase that removes the C-terminal pro-peptide of collagen I, promotes the production and assembly of fibrillar collagens. Elevated expression of BMP1 enhanced the cleavage of cancer-cell-derived procollagen and reduced PDAC tumor growth and metastasis, suggesting that accurate processing of collagen I also plays a key role in controlling PDAC progression [74]. Thus, collagen I heterotrimers produced by myCAFs act in a tumor-restrictive manner, whereas pancreatic cancer cell-expressed collagen I homotrimers promote PDAC tumorigenesis. This implicates a collagen I homotrimer-integrin α3β1 signaling axis as a cancer-specific therapeutic target [73].

## 7. ZEB1^+^ CAFs

The transcription repressor ZEB1 (Zinc finger E-box binding homeobox 1) becomes aberrantly expressed in human PDAC [75,76]. Several observations have identified ZEB1 as a regulator in pancreatic Kras-driven carcinogenesis, demonstrating that *Zeb1* deletion in the KPC mouse model significantly delayed the onset of pancreatic cancer with a remarkably low number of α-SMA^+^ fibroblasts [77,78]. *Zeb1* downregulation in mouse PSCs retarded the expansion of stromal myofibroblasts during precursor-to-cancer progression and interfered with their functions to foster the growth of pancreatic cancer cells [78]. Notably, *Zeb1* deficiency in PSCs prevented their transdifferentiation into myofibroblasts and reduced the expression of myCAF markers such as *α-SMA*, *Col1a1* and *Mmp9*, as was similarly shown for buccal mucosal fibroblasts [79]. Of functional relevance, ZEB1-expressing CAFs stimulated RAS activity in pancreatic cancer cells, supporting their migration, invasion and proliferation [78]. These findings support the view that ZEB1^+^-CAFs represent pro-tumorigenic myCAFs.

## 8. LRRC15^+^ CAFs

Stromal fibroblasts in many solid tumor entities express a membrane-bound protein called leucine-rich repeat containing 15 (LRRC15), which is involved in cell–cell and cell–ECM interactions [80,81]. LRRC15 overexpression in triple-negative breast cancer CAFs augments migration and invasion of cancer cells by inducing β-catenin expression and its nuclear localization [82]. With respect to PDAC, single cell RNA sequencing identified a subpopulation of CAFs expressing LRRC15 in mouse models and patients. This subtype is induced by TGF-β (and thus represents a myCAF subpopulation) and is absent in the pancreases of healthy subjects [50]. Of high importance is the observation that PDAC patients showing high expression of a LRRC15^+^ CAF signature responded poorly to anti-PD-L1 therapy. Another recent study confirmed this finding in mouse models, where the depletion of LRRC15^+^ CAFs led to a reduction in total fibroblast content of the tumor and a broad inactivation of fibroblasts [83]. As a result, cytotoxic T cells started to accumulate in the vicinity of the tumor, and anti-PD-L1 therapy became effective again. However, LRRC15^+^ CAFs reappeared after the initial ablation, underscoring the highly dynamic nature of the TME.

## 9. GLI1^+^ and HOXB6^+^ CAFs

As outlined above, CAFs might result from different cellular precursor pools. For most of the CAF populations described in this review, the cellular origins are not well defined. However, one recent study used lineage-tracing experiments to delineate the origins of Gli1^+^ mesenchymal cells in murine PDAC [84]. Gli1 represents a major target gene of the Hh signaling pathway and can thus be utilized as a read-out for pathway activity [85,86]. Apart from TGF-β, Hh signaling constitutes one of the central signaling systems activating pancreatic mesenchymal cells (fibroblasts and PSCs) in the tumor stroma [40,64,87,88]. Both TGF-β and Hh promote the formation of myofibroblastic (as opposed to inflammatory) CAFs, and these two pathways closely interact with each other on a mechanistic level [56,89,90,91]. With respect to lineage tracing, Gli1^+^ cells increased in number during carcinogenesis, while in contrast, Hoxb6^+^ (a marker of early pancreatic mesenchyme) cells did not. Furthermore, the former cells were enriched around blood vessels and pancreatic ducts, whereas the latter ones were dispersed throughout the stroma. Although a direct comparative assessment of these two subtypes was not made, the study by Garcia et al. strongly argues that distinct lineages exist in the pancreatic mesenchymal compartment which give rise to spatially and potentially also functionally discrete cellular populations in cancer [84]. These findings are in congruence with other data suggesting that PSCs might be a specific source of ECM-producing CAFs, but not for other CAF subtypes such as inflammatory CAFs [27]. In summary, the available data implies that two levels of CAF specification cues might exist: the developmentally determined cell subtype and the existence of local signaling molecules.

## 10. Inflammatory CAFs (iCAFs)

The inflammatory properties of CAFs across multiple cancer types have been noted in numerous studies. In PDAC, the explicit categorization into inflammatory iCAFs and myofibroblastic myCAFs, initially described by Tuveson and colleagues, is now broadly established [49,92,93]. Inflammatory CAFs are located at a considerable distance from malignant cells and express extremely low levels of α-SMA, but produce inflammatory cytokines such as IL-6, leukemia inhibitory factor (LIF), SAA1/SAA3 and CXCL1 [47,49,94]. Many, if not all, of these cytokines play important roles in shaping the tumor immune microenvironment. For instance, IL-6 is an inflammatory cytokine secreted by iCAFs in response to paracrine signaling from the malignant epithelium, which leads to enhanced invasion and colony formation of pancreatic cancer cells [49,95]. Elevated IL-6 levels in human serum are an independent risk factor for the development of extensive hepatic metastases in patients with PDAC [96]. IL-6 and numerous other pro-inflammatory genes were shown to be highly expressed in unsorted primary human PDAC CAFs [97]. CAF-derived IL-6 is a major contributor to immune evasion in PDAC [98]. Additionally, CAF-secreted IL-6 increased glycolytic flux in pancreatic tumor cells and lactate efflux in the microenvironment, favoring the activation of M2 macrophages in the TME and excluding CD8^+^ T cells [99]. In the KPC mouse model of PDAC, iCAF-derived IL-6 travels to the liver in an endocrine manner and induces STAT3 signaling in hepatocytes, forming the pre-metastatic niche for PDAC liver metastases [100]. Treatment of KPC mice with a monoclonal antibody against IL-6R in combination with gemcitabine resulted in tumor regression and ultimately increased animal survival [101]. Furthermore, a combined IL-6/PD-L1 blockade resulted in decreased tumor volumes in a syngeneic orthotopic PDAC model in a CD8^+^ T cell-dependent manner [97]. Despite numerous positive pre-clinical data proving the beneficial outcome of IL-6 targeting in PDAC, the complete lack of IL-6 in the TME of PDAC had surprisingly no significant impact on the development of pancreatic cancer in mouse models. Nevertheless, IL-6 deletion in α-SMA+ CAFs improved gemcitabine efficacy and synergized with ICB [102]. Suppression of IL-6 likely favors the emergence of effector T cells, which, when combined with the gemcitabine-induced cell death and the generation of neoantigens, collaborate with CD11c^+^ dendritic cells, augmenting the efficacy of the immune checkpoint blockade [102].

Another critical factor released by iCAFs is Leukemia inhibitory factor (LIF), a key paracrine mediator [94,103]. In both mouse and human PDAC tissues, *LIF* mRNA was abundant in specifically activated CAFs, especially in those adjacent to cancer cell nests, reaffirming activated PSCs as the major LIF-producing cells. Genetic ablation of *LIFR* and pharmacologic LIF blockade in the KPC mouse model demonstrated that LIF mainly acts on pancreatic cancer cells to facilitate tumor progression but not initiation. These approaches targeting LIF augmented the efficacy of chemotherapy to prolong survival of PDAC mouse models, primarily by modulating cancer cell differentiation and epithelial–mesenchymal transition status [103]. Similarly, LIF depletion by genetic means or by neutralizing antibodies prevented engraftment in pancreatic xenograft models and synergized with gemcitabine to eradicate established pancreatic tumors in a syngeneic, Kras^G12D^-driven PDAC mouse model [104]. Moreover, the secretome of pro-tumorigenic IL-17A-producing CD8^+^ T cells (Tc17 cells) enhanced *Lif* expression in PSCs, establishing Lif as a genuine iCAF marker [105]. Notably, neuropathic pain is a major clinical problem caused by PDAC-associated neuronal remodeling (PANR). It was shown that LIF, secreted by PDAC stromal cells, induced not only nerve cell migration, but also differentiation, thereby positively correlating with PANR and axonogenesis [106]. A LIF-blocking antibody reduced intratumoral nerve density in PDAC mice, supporting a critical role for LIF targeting to limit PANR in PDAC pathophysiology. Taken together, LIF represents an attractive therapeutic target and biomarker in PDAC and warrants further comprehensive evaluation for clinical application.

With respect to identified markers, the expression of PDGFRα has been proposed as a surface marker of iCAFs [47,107], which is in line with PDGFRα-inhibiting myCAF differentiation through MRTF-dependent mechanisms [108]. Other iCAF sub-populations such as CXCR4^+^ and CD133^+^ iCAFs have also been identified in single-cell approaches, but their pathophysiological significance is currently unresolved [109]. In general, inflammatory CAFs display marked NF-κB and JAK-STAT3 signaling as a result of IL-1-mediated paracrine activity [94]. IL-1 signaling is the main pathway responsible for the induction of an inflammatory phenotype in CAFs. IL-1α activates JAK/STAT signaling pathways in PSC via STAT3 and induces expression of IL-1R [94]. Therefore, phosphorylation of STAT3 was suggested as a surrogate marker of activated iCAFs but not myCAFs. Interestingly, LIF has been identified as an autocrine factor secreted by IL-1α-activated PSCs and plays a major role in the amplification of JAK/STAT signaling in iCAFs. JAK Inhibition shifts iCAFs to a myofibroblastic phenotype in vivo, suggesting that inhibitors of the IL-1/JAK/STAT pathway are a potential anti-cancer strategy [94]. Furthermore, disruption of the STAT3 signaling axis via genetic ablation of *Stat3* in Fsp^+^ stromal fibroblasts in the KPC PDAC mouse model not only slows tumor progression and increases survival, but also reshapes the characteristic immune-suppressive TME by decreasing M2 macrophages (F480^+^CD206^+^) and increasing cytotoxic CD8^+^ T cells [110]. In addition to IL-1, IL-17A and TNFα produced by CD8^+^, T cells can also induce iCAFs [105]. Blocking IL-1R or IL-17R signaling in pancreatic fibroblasts or stellate cells has been shown to be sufficient to reduce PDAC progression [94,105,111], offering potential pharmacological approaches targeting IL-1α, IL-β, PDGFRα or IL-17A pathways for PDAC interventions (e.g., NCT03086369; NCT02021422). Translationally important, chemoresistant patient samples depicted a higher content of iCAFs than myCAFs, suggesting a functional role for inflammatory CAFs in chemoresistance [109].

## 11. SAA3^+^ CAFs

Serum amyloid A3 (SAA3), a member of the serum amyloid A apolipoprotein family, has been found to be associated with high-density lipoproteins in plasma, which are observed during chronic inflammation and cancer, and to correlate with reduced survival in human cancer [112]. SAA1 and SAA3 are upregulated in inflammatory CAFs [49] and also in apCAFs [47]. Mariano Barbacid and colleagues characterized the transcriptome of a protumorigenic subpopulation of CAFs defined by the expression of PDGFRα and identified that Saa3 was markedly upregulated in PDGFRα^+^ iCAFs, regulating their functions [107]. The authors demonstrated that PDGFRα^+^, but not normal pancreatic fibroblasts, promoted tumor growth in vitro and in vivo. Conversely, *Saa3*-null PDGFRα^+^ CAFs inhibited tumor growth. However, this antitumor phenotype was abrogated when *Saa3* was knocked down in both tumor cells and CAFs, making targeting of SAA3 potentially challenging in clinical practice. Nonetheless, PDGFRα might be a viable therapeutic target against iCAFs by using the PDGFRα monoclonal antibody olaratumab for metastatic PDAC (NCT03086369).

## 12. Complement-Secreting CAFs (CsCAFs)

Using single cell transcriptomics, Chen and colleagues were able to identify a new subpopulation of CAFs in the proximity of malignant ductal cells of early-stage PDAC [113]. These cells expressed many components of the complement system such as C3, C7, CFB, CFD, CFH and CFI; hence, they named these cells complement-secreting CAFs (csCAFs). Potentially, csCAFs may play a role in modulating immune and inflammation responses, but further studies are still needed to reliably link these cells to specific functional processes in PDAC.

## 13. FAP^+^/CXCL12^+^ CAFs

Fibroblast activation protein α (FAP) is a serine protease with dipeptidyl peptidase and gelatinase activity that is expressed by CAFs in over 90 percent of epithelial cancers, including pancreatic cancer [114,115,116,117]. Intriguingly, depleting FAP^+^ stromal cells in a subcutaneous PDAC mouse model rendered these tumors sensitive to immune checkpoint therapy, which is commonly ineffective in PDAC [118]. Mechanistically, FAP^+^ CAFs secrete the C-X-C motif chemokine 12 (CXCL12), which inhibits intratumoral accumulation of cytotoxic CD8^+^ T cells, thereby promoting an immunosuppressed microenvironment. Blockade of CXCR4, the CXCL12 receptor, in the KPC mouse model not only leads to a rapid accumulation of CD8^+^ T cells in the vicinity of cancer cells but also enhances the effectiveness of anti-PD-L1 therapy [119]. In human breast cancer, CAF-derived CXCL12 promotes tumor growth and angiogenesis [120]. Specifically, mammary myofibroblastic CAFs (termed CAF-S1) attract regulatory and immune-suppressive CD4^+^CD25^+^ T lymphocytes in a CXCL12-dependent manner [53]. In addition, in metastatic breast cancer, inhibition of CXCR4 in CAFs leads to an increase in T cell infiltration and improved effectiveness of immunotherapy [121]. Another study demonstrated that FAP from CAFs cleaves type I collagen, which enhances the adhesion of tumor-associated macrophages through class A scavenger receptors [122]. Collectively, these findings suggest that FAP^+^ CAFs use different mechanisms to generate an immunosuppressive microenvironment. In addition, overexpression of FAP in fibroblasts leads to an alteration of the extracellular matrix and enhances the invasiveness of pancreatic cancer cells [123]. In the KPC mouse model, FAP^+^ CAFs support not only metastasis but also the progression of PDAC [124]. Moreover, higher expression of FAP in CAFs of PDAC patients is associated with a worse clinical outcome [124,125], suggesting FAP and/or CXCL12 as promising therapeutic targets.

## 14. CD10^+^/GPR77^+^ CAFs

From a study of multiple cohorts of breast and lung cancer, a new CAF subtype was identified: Fibroblasts expressing elevated levels of the two surface markers CD10 and GPR77 [126]. CAF-positivity for CD10 was well described in pancreatic cancer, but their functionality has not been assessed in detail [127,128]. In mammary and pulmonary malignancies, CD10^+^/GPR77^+^ CAFs correlated with chemoresistance and poor patient survival [126]. In this study, the authors propose a model in which CD10^+^GPR77^+^ CAFs provide the conditions to maintain a niche for cancer stem cells (CSCs) through the release of IL-6 and IL-8. This is mediated by sustained NF-κB signaling through phosphorylation and acetylation of the p65 subunit [126], thus increasing transcriptional NF-κB activity [129,130]. NF-κB signaling is maintained active through a positive feedback loop: NF-κB promotes the release of complement factor C5a, a GPR77 ligand. Upon binding of C5a, GPR77 phosphorylates and activates RSK1, which in turn mediates phosphorylation of the p65 subunit of NF-κB. Furthermore, treatment with neutralizing anti-GPR77 antibodies in mice with xenografts restored chemosensitivity and reduced tumor proliferation [126]. However, complement-secreting CAFs (csCAFs, see specific paragraph) have been identified specifically in early but not late PDAC, suggesting that they might play a role in tumor restraint [113]. It is currently unclear whether the reasons for this discrepancy are due to different malignancies or different experimental set-ups.

A subsequent study on breast cancer focused on the role of stromal CD10. CD10 is a transmembrane metallopeptidase that, according to Yu et al., helps to maintain tumor cell stemness by degrading the extracellular osteogenic growth peptide (OGP) [131]. In a paracrine manner, OGP suppresses the expression of the desaturase SCD1, an enzyme required for the desaturation of lipids thought to be essential for CSCs [132]. Furthermore, experiments in mammary xenograft mouse models have shown that silencing CD10 in CD10^+^GPR77^+^ CAFs restored chemotherapy efficacy [131]. CD10^+^GPR77^+^ CAFs proved to have a relevant role also in other cancer entities. For example, elevated CD10, GPR77 and fibroblast activation protein-α (FAP) expression was used as a predictive biomarker in gastric cancer patients and was found to be associated with chemoresistance and poor prognosis [133].

## 15. Hypoxia^+^ CAFs

CAFs are known to modulate the immune response and inflammation in the TME, and hypoxic CAFs exert these effects by releasing an array of chemokines, cytokines and signaling factors. For example, in prostate cancer CAFs, hypoxia-activated HIF-1 induces transcription of TGF-β, which in turn promotes the release of CXCL13. This chemokine attracts B lymphocytes and promotes tumor progression and invasiveness [134]. Similarly, melanoma-hypoxic CAFs can inhibit T cell activity via the release of immunosuppressive molecules such as TGF-β, IL-6, IL-10, VEGF and PD-L1 [135]. Furthermore, hypoxia has been found to stimulate transcription of the CXCL12 receptor (CXCR4) and the IL-6 receptor in different cell lines including mouse embryonic fibroblasts and prostate CAFs [136,137]. Hypoxia can suppress immune activity by making the TME unfavorable to T cells: CAFs extracted from PDAC tissue and exposed to hypoxia conditions over-express arginase II (ARG2), an enzyme that converts arginine, an amino acid essential for T cell activity, into ornithine [138].

It has been reported that hypoxia can drive CAFs toward an inflammatory phenotype via induction of IL-1α [139,140]. Although the vast majority of studies focus on HIF-1, HIF-2 also plays an important role in the crosstalk between tumor cells and CAFs. An in vivo study on pancreatic cancer TME showed that deletion of *Hif2a* but not *Hif1a* in CAFs decreased immunosuppressive M2 macrophages and regulatory T cells infiltration. Further in vivo experiments using a HIF2 inhibitor increased immunotherapy efficacy [141].

Although there appears to be a consensus on the tumor-promoting influence of hypoxia, the role of HIF-1 in the TME is less clear. Interestingly, in an in vivo study by Kim et al., fibroblast-specific deletion of *Hif1a* increased mammary cancer cell growth and tumor perfusion, but reduced infiltration of tumor-associated macrophages. Knockout of *VEGFA* in CAFs had similar effects [142]. Other studies on human head and neck and vulval CAFs showed that, strikingly, hypoxia reverses CAF activation and decreases ECM remodeling, reducing cancer cell invasiveness. Similar results were obtained by chemically inhibiting the HIF-1 inhibitors PHDs [143].

## 16. Antigen-Presenting CAFs (apCAFs)

In 2019, David Tuveson and colleagues not only described myCAFs and iCAFs in PDAC, but they also identified a third CAF subtype expressing MHC class II (MHC-II) genes and CD74 [47]. The authors termed this cell population *antigen-presenting CAFs* (apCAFs) and found that they induce T cell receptor (TCR) ligation in CD4^+^ T cells in an antigen-dependent manner, confirming their putative immune-modulatory capacity. However, apCAFs lack the costimulatory molecules needed to induce T cell proliferation. Moreover, a negative correlation between the abundance of these apCAFs and the ratio of effector CD8^+^ T cells and regulatory T cells (Tregs) was observed in human PDAC. Therefore, it has been hypothesized that MHC-II expressed by apCAFs acts as a decoy receptor to deactivate CD4^+^ T cells by inducing either anergy or differentiation into Tregs, contributing to immune suppression in the PDAC TME [47,144]. Surprisingly, apCAFs were not detected in co-culture models, demonstrating exclusive polarization of PSCs into myCAFs and iCAFs [49]. This suggests that apCAFs might not originate from PSCs. Instead, mesothelial cells have been described as cells of origin of apCAFs due to their overlapping gene expression signatures and antigen-presenting capacity [50]. Indeed, a later study demonstrated that apCAFs are derived from mesothelial cells [144]. They found that, during late stages of PDAC, mesothelial cells maintained the expression of MHC-II genes, gained activated CAF markers (e.g., α-SMA or IL-6) and decreased the expression of certain mesothelial markers (e.g., mesothelin (MSLN)). Antigen-presenting CAF-specific gene signatures were mainly driven by NF-kB and TGF-β signaling, and intriguingly, both IL-1 and TGF-β can induce a mesothelial-to-CAF transition [144]. Of therapeutic interest, MSLN-targeting antibodies could significantly prevent or reduce this transition, resulting in fewer Tregs and higher numbers of cytotoxic T cells in mouse model tumors. Therefore, targeting MSLN may be a potential strategy to effectively inhibit apCAF-dependent immunosuppressive features.

## 17. CD105^+^ CAFs

In the context of lineage tracing, CD105-associated CAFs have gained increased attention. Hutton et al. utilized sophisticated sorting techniques to identify distinct fibroblast lineages in healthy or cancerous tissue, which could be distinguished according to the presence of the transmembrane glycoprotein CD105 (also known as endoglin). In most analyzed PDA samples, CD105^pos^ CAFs were more abundant, with a 7:3 ratio, although considerable variability was observed [145]. It was found that both CD105^pos^ and CD105^neg^ lineages could polarize into myCAFs or iCAFs. However, apCAF polarization was exclusively present in CD105^neg^ CAFs, as indicated by the apCAF gene expression restricted to the End^neg^ cluster [145]. In concordance with recent findings that apCAFs are derived from mesothelial cells [144], it was observed that a subset of CD105^neg^ CAFs expresses higher levels of mesothelial cell-associated genes [145]. Another intriguing finding of CD105^pos^ and CD105^neg^ CAFs was their opposing effect on several immune subsets. Only CD105^neg^ CAFs were able to stimulate adaptive immunity with regard to T cell and general DC infiltration. Thus, murine models showed a drastic restriction in tumor growth when pancreatic tumor cells were co-injected with CD105^neg^ CAFs. In contrast, co-injection of cancer cells with CD105^pos^ CAFs resulted in accelerated tumor growth. Interestingly, even when co-injected in a 1:1 ratio with CD105^pos^ CAFs, CD105^neg^ CAFs were able to suppress tumor growth, although to a lesser extent. Mechanistically, various signaling pathways such as TGF-β/NFκB, as well as TNF-α/IL-6, were enriched in CD105^pos^ or CD105^neg^ CAFs, respectively [145].

In colorectal cancer, CAF-specific CD105 expression is associated with increased invasion and CAF survival [146]. The CD105-neutralizing antibody TRC105 (also known as Carotuximab) [147,148] reduced Smad1 phosphorylation in CAFs and decreased fibroblast-mediated liver metastases [146]. However, in the context of pancreatic cancer, targeting CD105^pos^ CAFs could not inhibit tumor growth in the KPC model [149]. Although an increased percentage of tumor-infiltrating CD8^+^ cytotoxic T cells was observed after TRC105 treatment, this did not translate into changes in the tumor volume or mass. Furthermore, the combination of TRC105 and an α-PD-1 antibody therapies could not inhibit tumor growth in KPC [149]. These findings illustrate that, despite being abundantly present in the pancreatic tumor stroma, CD105^pos^ CAFs are not exclusively responsible for the tumor-promoting effects of CAFs. It is also plausible that CD105 delineates a broader CAF population, which may need to be subdivided into more specific subsets according to their contribution to pancreatic cancer progression.

## 18. Meflin^+^ CAFs

Coincidentally, in 2016, Maeda et al. identified a novel cell surface protein on MSCs, pericytes and fibroblasts which they termed Meflin [150]. They found that Meflin maintains an undifferentiated state in cells expressing the surface protein and is downregulated upon differentiation [150]. The subpopulation of Meflin^+^ CAFs seems to harbor cancer-suppressive properties, and Mizutani et al. were able to detect Meflin on a small subpopulation of α-SMA^low^ FAP^+/−^ PDGFRα^+^ Gli1^+^ CAFs [151]. Moreover, alterations in Meflin expression in MSCs were able to induce gene expression changes associated with CAF activation. For example, Meflin depletion in MSCs resulted in upregulation of α-SMA and Gli1 as well as cytokines IL-6 and CCL2, displaying an activated CAF gene signature [151]. Xenograft models with Meflin-KO mice exhibited a higher tumor burden [151]. The mechanism by which this CAF subpopulation restricts tumor progression was found to be associated with a favorable response to immune checkpoint blockade (ICB) therapy. In non-small cell lung cancer (NSCLC), patients with increased numbers of Meflin-positive CAFs exhibited greater tumor vessel areas and higher numbers of infiltrating CD4^+^ T cells in the stroma [152]. In PDAC transplantation mouse models, Meflin expression induced by administering Am80 (a structural analog of the retinoid acid receptor (RAR) α-selective agonist Am580) showed increased chemosensitivity to gemcitabine, attenuation of tissue stiffening by inhibition of lysyl oxidase and increased tumor vessel area [153]. These observations suggest a tumor-suppressive phenotype by ensuring an enhanced drug delivery to the tumor site facilitated by increased vascularization and abrogation of tissue stiffening. Moreover, estimating the proportion of Meflin-positive CAFs in a tumor could be utilized to estimate a favorable response to chemosensitivity.

## 19. CD271/NGFR^+^ CAFs

CD271, also known as NGFR or p75, is a member of the TNF receptor family and was found to be expressed on activated HSCs and PSCs [154]. As CD271 harbors a death domain, stimulation with the CD271 ligand NGF was able to induce apoptosis in HSCs [154]. These findings initiated a study by Fujiwara et al., in which a significant correlation between high numbers of CD271-positive CAFs and a better prognosis of PDAC patients was observed [155]. Co-culture of PSCs and PCCs or stimulation by PCC-conditioned medium increased expression of CD271 in PSCs, but only transiently. Thus, it was proposed that CD271 could serve as a temporary marker for PSCs in the early stages of tumor development [155]. Interestingly, CD271-positive CAFs were found to be located more distantly from pancreatic tumor cells compared to their CD71-negative counterparts [127,155]. In contrast, other studies which focused on CD271-positive BM-MSCs (possible precursors of CAFs) in gastric cancer revealed CD271-positive stromal cells to be associated with worse outcomes [156,157]. Therefore, more work is still needed to assess the impact of CD271 as a reliable prognostic marker.

## 20. Metabolic CAFs

It is known that tumor cells undergo a series of reprogramming to meet the metabolic requirements needed to keep up with the elevated biomass production. One of the most famous of these changes is the Warburg effect, a phenomenon in which cancer cells preferentially use the glycolytic pathway rather than the more efficient oxidative phosphorylation for energy production, even in the presence of oxygen [158]. However, metabolic reprogramming is not a feature exclusive to tumor cells. An increasing number of studies are pointing to metabolic changes in CAFs and how these can influence tumor progression. Contrasting other CAF subtypes described in this review, which can be defined by specific marker genes, metabolic adaptations are highly plastic in CAFs and may depend on the specific local microenvironment being investigated. As such, metabolic changes might apply to many of the “marker-defined” CAF subsets outlined in this work.

In a 2009 study, Pavlides et al. introduced the concept of the “Reverse Warburg Effect”, whereby tumor cells induce metabolic changes in CAFs, forcing them to secrete energy-rich molecules such as pyruvate and lactate, even under normoxic conditions [159]. The secretion products of these glycolytic CAFs can subsequently be used by cancer cells to support growth [159,160]. Proteomic analysis showed that the loss of caveolin-I (Cav-I; a TGF-β receptor inhibitor) resulted in an upregulation of glycolytic enzymes. Furthermore, they found the loss of stromal Cav-I was associated with a more aggressive breast cancer behavior [159]. Similarly, in a pancreatic cancer study, CAFs were found to overexpress the enzymes PKM2 and LDHA and to show increased glucose uptake and lactate production/secretion through overexpression of the MCT1/4 lactate transporters, enhancing cancer cell invasiveness [161]. Among the main factors that can induce such metabolic changes are TGF-β ligands released by tumor cells and the activation of HIF-1α. For example, TGF-β1 or PDGF can induce the activation into myCAFs and iCAFs, respectively, and promote a switch from oxidative phosphorylation to aerobic glycolysis, as indicated by miRNA-mediated downregulation of isocitrate dehydrogenase 3α (IDH3α) [162]. The reduction in IDH3α provokes a series of metabolite imbalances that cause the inhibition of PHD2, HIF-1α protein stabilization and the subsequent transcription of glycolysis-related genes [162].

CAFs can also metabolically support tumor cell progression by providing amino acids. Several studies have shown that CAFs can synthesize glutamine, an important carbon and nitrogen source, and release it into the TME to benefit cancer cell proliferation [163,164]. Similarly, PSCs can fuel tumor cell growth by releasing alanine in an autophagy-dependent manner, providing cancer cells with an alternative source of carbon [165]. In general, stromal autophagy has now been recognized as an important paracrine mechanism in several malignancies [166,167,168,169], and elevated ECM rigidity (as occurs through activated myCAFs) seems to promote this process even further [58]. Furthermore, and as mentioned above, lactate is capable of suppressing p62 in the tumor stroma [26], resulting not only in CAF activation, but also in an ATF4-driven pro-tumorigenic asparagine secretion into the TME [170]. Moreover, CAFs can even migrate toward glutamine-richer areas, which in turn can induce the invasion of tumor cells [171]. Taken together, CAFs are critical metabolic players in the pancreatic TME.

## 21. Stromal Targeting: Where Do We Stand?

Given the abundance of stroma in PDAC, CAFs have been the focus of therapeutic targeting for an extended period [172]. However, due to the presence of tumor-promoting as well as tumor-restraining CAF sub-populations, therapeutic approaches need to become more selective than initially anticipated. Moreover, an in-depth understanding of the multitude of interactions taking place between cancer cells, mesenchymal cells and immune cells will be paramount. Furthermore, CAFs are highly plastic cell types and can dynamically transition into other subpopulations as a result of therapy-induced alterations in extracellular ligand concentrations or ECM composition [94,173]. In that respect, targeting the paracrine Hh signaling pathway can serve as a negative example due to the elimination of large parts of the myofibroblastic stroma and the unexpected subsequent deterioration of survival in mice and humans [56,174].

A comparable clinical disappointment was observed in the targeting of hyaluronan (hyaluronic acid (HA)), which is produced preferentially by myCAFs and represents the second most abundant component of ECM produced by CAFs. HA attracts and retains water molecules, resulting in elevated interstitial fluid pressure. Supraphysiological HA levels in tumors lead to an outward interstitial fluid flow and the compression of blood vessels, making the delivery of chemotherapeutic drugs to the tumor highly inefficient. Therefore, HA has been classified as a promising anti-cancer target, as was also verified in several preclinical models [41,42,72,175]. Removing HA using a PEGylated form of the HA-degrading enzyme hyaluronidase (PEGPH20) proved promising in pre-clinical testing and early clinical trials, but eventually failed in phase III [175,176,177]. The underlying reasons for this discrepancy are currently unresolved, but might be linked to the pro-inflammatory effects of the degraded low-molecular-weight hyaluronic acid or different functions of high and low molecular weight hyaluronan [178,179,180] These findings suggest the consideration of other inhibitors of HA-induced signals, including 4-methylumbelliferone, which has been successfully tested in pre-clinical models of lung and prostate cancer [181,182].

An alternative approach being clinically pursued is the induced transition of activated CAFs into quiescence by Vitamin D receptor (VDR) agonists (e.g., Calcipotriol) or retinoic acid receptor agonists (e.g., all-trans retinoic acid (ATRA)) [23,183]. Both treatments reverse the activation of CAFs, render cells mechanically and metabolically quiescent and re-induce the formation of lipid droplets, which are characteristic of non-activated stellate cells in the liver and pancreas. ATRA has been evaluated as safe in a phase I trial (NCT03307148) and will be pursued as a combination therapy together with Gemcitabine/Nab-Paclitaxel in a phase II randomized trial [184]. With respect to VDR stimulation, the synthetic agonist Paricalcitol is currently being tested in several pancreatic cancer clinical trials in combination with chemotherapy (e.g., NCT04617067, NCT03883919, NCT03472833, NCT03520790, NCT03519308).

Another angle to tackle tumorigenesis might be the “normalization” of the altered biophysical properties encountered in tumors. Due to the vast deposition of ECM, PDAC tissue is notoriously stiff, promoting cancer cell dissemination through induction of EMT and blocking the infiltration of cytotoxic CD8^+^ immune cells as well as the pro-tumorigenic metabolic rewiring of cancer and stroma cells [58,61,185,186,187,188,189]. Overall, these changes result in a general increase in tumor aggressiveness, and targeting the mechanobiology of the stroma is an intriguing concept in stromal therapy. One such approach could be the inhibition of the Rho-associated kinases ROCK1/2. ROCK inhibition has been shown to enhance chemosensitivity, reduce metastasis and improve outcomes in preclinical PDAC models [59,62]. In addition, small-molecule inhibition of focal-adhesion kinase (FAK) reduced fibrosis and the infiltration of immune-suppressive immune cells in PDAC mouse models [190]. Most importantly, targeting this mechano-transducing kinase rendered tumors sensitive to immune checkpoint blockade and significantly extended the survival of treated animals. However, it should be mentioned that these findings contrast with another study that reported on the control of pro-tumorigenic cancer cell glycolysis by CAF-localized FAK [18]. Along with the latter findings, a recent phase II clinical trial of a FAK inhibitor in conjunction with MEK blockade was not very successful (NCT02428270; [191]). However, several additional trials employing FAK inhibition in PDAC are currently ongoing (e.g., NCT05580445, NCT00666926, NCT02546531, NCT03727880), and it will be interesting to follow up on their results. In addition to the aforementioned pharmacological strategies, biophysical approaches (e.g., pulsed focused ultrasound) aiming to improve tumor tissue mechanics have yielded promising results by increasing immune cell invasion [192].

Another potential treatment might be the use of the anti-fibrotic drug Losartan, an antagonist of the angiotensin II receptor AT1. This drug has long been used to lower blood pressure and is clinically and toxicologically well-established. Losartan reduces fibrosis and solid stress in PDAC tissues, improves drug delivery and potentiating chemotherapy in preclinical models [193,194]. Clinical trials are ongoing (NCT01276613, NCT04106856, NCT01821729), and Losartan might show benefits in downstaging locally advanced patients in conjunction with neoadjuvant radiochemotherapy [195,196]

Recent evidence surprisingly revealed that Tamoxifen, a standard medication in the treatment of breast cancer and therefore well-established in the clinic, exerts mechanical reprogramming activity in the PDAC stroma through its effects on CAFs/PSCs [197,198]. Mechanistically, it acts through G-protein-coupled estrogen receptors (GPERs) and not through classical nuclear estrogen receptors. The promising preclinical results from Tamoxifen-induced stromal targeting have so far not been validated in the clinical setting, but this will certainly be worth testing. Of interest could also be the endogenous bioactive lipid Lipoxin A_4_ (LXA4), which functions as a natural antagonist of TGF-β, fibrosis and dysplasia [199,200]. Lipoxins are inflammation-resolving mediators typically secreted by macrophages or neutrophils in response to infection or tissue injury. These molecules are synthesized from arachidonic acid and are known to bind to the GPCR formyl peptide receptor 2 (FPR2) [201]. LXA4 exerts anti-tumorigenic effects on CAFs/PSCs and macrophages, and due to its endogenous nature, it could potentially be harnessed as biological therapy in the future [202,203,204].

In conclusion, although stromal targeting has reached clinical testing, it is still far from being part of any therapeutic routine. Moreover, a number of setbacks have demonstrated that the multicellular TME is still not fully understood, and unexpected results can occur. Nevertheless, therapeutic remodeling of the CAF stroma may provide novel avenues for combination therapy that were previously ineffective.

## Figures and Tables

**Figure 1 cancers-15-02614-f001:**
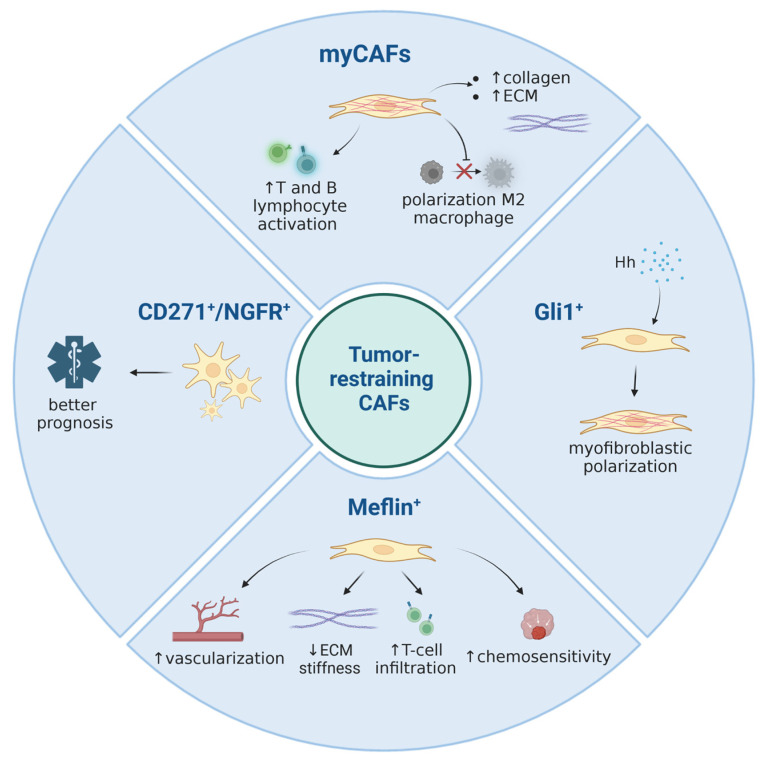
A summary of currently known tumor-restraining CAF subpopulations. Image created with Biorender.com.

**Figure 2 cancers-15-02614-f002:**
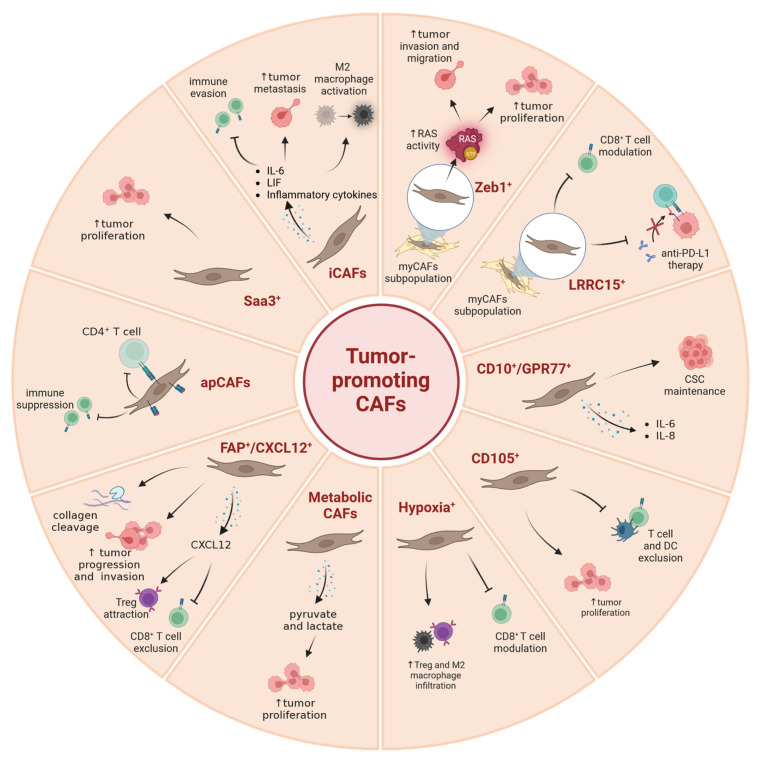
A summary of currently known tumor-promoting CAF subpopulations. Image created with Biorender.com.

**Figure 3 cancers-15-02614-f003:**
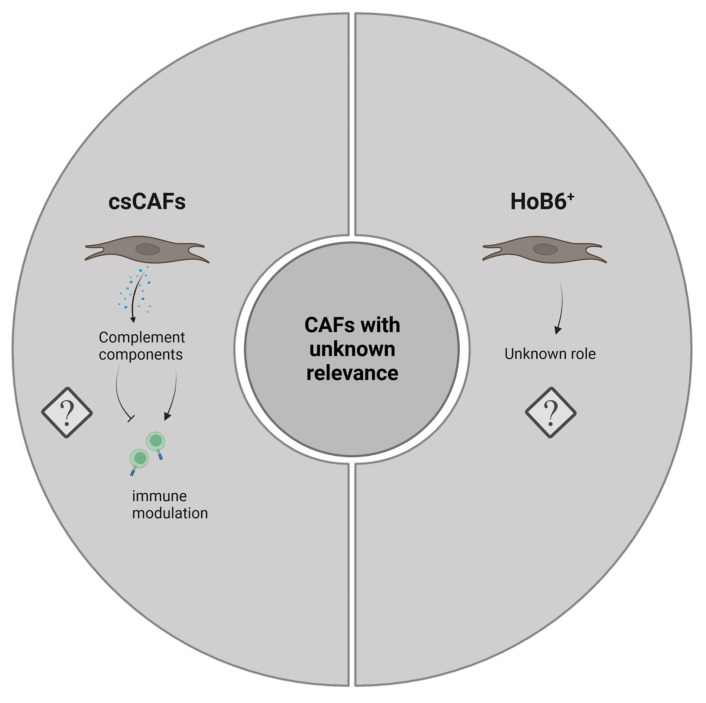
Graphical representation of CAF subtypes with currently unknown functionality in PDAC. Image created with Biorender.com.

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
