# Peer review of "A Quick Guide to CAF Subtypes in Pancreatic Cancer"

_cancers, 2023, doi:10.3390/cancers15092614_

Round 1
Reviewer 1 Report
In the manuscript “A quick guide to CAF subtypes in pancreatic cancer”, Brichkina et al. make an extensive compilation of different types of cancer associated fibroblasts in pancreatic cancer and their different roles.
Here I include some suggestions in order to make this guide the authors propose more comprehensive:
-Organization of the different CAF subtypes: As the authors write, two main type of CAFs have been described in PDAC: iCAF and myCAF. Zeb1+ and LRRC15+ fibroblasts seem to be myCAF subtypes, as they state in Fig. 2. It would be clearer if those subtypes were included within the myCAF chapter.
-Additionally, for other subtypes, such as Hypoxia + CAFs, are they different types of CAFs, or CAFs that change to adapt to the microenvironment?. I suggest to include them in a different chapter which could be named “Tumour-CAFs crosstalk” or similar, where “metabolic CAFs” could also fit. It would be interesting to extend a bit more on this topic and how CAFs respond to tumour (pancreas-derived cells) signalling, as the authors do on page 10 for example and specify IL1 secretion by PDAC tumour cells promotes the formation of iCAFs from PSC (Biff et al., Cancer Discovery 2019) (page 10).
-Since this is a very active field in pancreatic cancer with important implications as the authors state in "Stromal targeting: where do we stand?" I suggest to extend the discussion on CAFs plasticity and interconversion (Biffi et al., as before, Avery et al., Matrix Biology 2018...)
Minor comment:
-In page 11, line 414, the heading Complement-secreting CAFs appears with a different font compared to other headings.
Author Response
"Please see attached file"

Reviewer 2 Report
This is a very thorough review of the available work in the field of CAFs. The presentation is well organized and nicely highlights all the great developments that have been published but also how far we are from taking advantage of this this tumor cell population.
I would only suggest the authors add the multiple trial efforts with VDR, as that is missing.
Reviewer 3 Report
This is a good review paper. Please find some specific comments below.
Lane 11: Please delete “actually”.
Lane 15-16: Please rephrase the sentence to state clearly the objective of this review.
Lane 17: keywords are missing.
Lanes 21-24: Please rephrase, not scientific enough.
Lane 38: I am not sure what this sentence means, could make it clearer.
Lane 39-41: A reference is needed here.
Lane 42: Which scenario? Please be mor specific.
Lane 44: Please replace the word “ now”.
Lane 53: A reference needed ?
Lane 55: Please delete “an”.
Lane 56: Please rephrase “enabler”.
Lane 65: Review “probably” and please be more specific. Do they apply or not?
Lane 70: Why a capital letter on integrin and not eg. in actin or other proteins? Please be consistent throughout the manuscript.
Lane 80. “also” is used too many times. Please review and use only when necessary.
Lane 105: Delete “A few words on”. Not needed.
Lane 123 -127: Acronyms like PDAC, CAF, PSC should have been defined earlier in the manuscript.
Lane 175-660: This section is very well written.
Lane 320: Needs a more recent reference.
Lane 662: Maybe better to delete “in” and replace “for” with “of”?
Lane 672: I would avoid this negativism and oversimplification. In some cases HA approaches have worked well in cases with high MW HA
Lane 705: Should include biophysical approaches for targeting PDAC tumours eg. use of focused ultrasound, example doi: 10.1098/rsif.2021.0266.
Round 2
Reviewer 3 Report
The authors have responded to all my suggestions in full.